# Making Bedlam: Toward a Trauma-Informed Mad Feminist Literary Theory and Praxis

Jessica Lowell Mason

Department of Global Gender and Sexuality Studies, University of Buffalo, Buffalo, NY 14260, USA; jlmason1@buffalo.edu

**Abstract:** Building on what Margaret Price describes as the "long history of positive and person-centered discourses" of the term Mad, this article seeks to offer a (re)tooling and (re)theorization of the not-so-antiquated concept of "bedlam" as part of a Mad feminist literary theory and practice that aims to situate reading and writing practices on the subject of madness within a trauma-informed Mad framework and to (re)shape reading and writing practices by (re)seeing or seeing-in-a-new-and-old-Mad way the concept of "bedlam"—rendering it agential and unhinging it from its historical meanings. The article theorizes "bedlam" as a form of deliberate Mad literary practice, offering two examples of "bedlam-making", one in the poetry of Anne Sexton's 1960 collection *To Bedlam and Part-way Back* and the other in the historical fiction of Toni Morrison's *Beloved*. The article strives to re-articulate "bedlam" in a way that draws attention to the agency of language on the subject of madness, when written and read by writers and readers aware of the acute violences and traumas performed upon bodies exiled from "Reason", attending to the ways in which writers and readers make a subjectivity of "bedlam" or a resistance to and critique of systemic oppression that gives social agency to Mad literary action. "Making bedlam", it is argued in this essay, is a Mad feminist literary theory and practice, part of social justice discourses and liberation-focused action, which is deeply connected with other liberation movements in pursuit of the end of systemic violences.

**Keywords:** *Beloved*; Toni Morrison; poetry; Anne Sexton; trauma; madness; Mad; mental illness; disability; Mad studies; colonialism; violence; bedlam; gender; race; feminist theory

## 1. Introduction: Bedlam as Mad Feminist Literary Theory

The word "bedlam" may bring to mind 18th-century images of naked figures bound and chained in public squares or dungeon-like quarters, prostrate bodies stretched in rebellion or submission, in agony, covered and surrounded by filth. It may bring to mind periodized images of dormitory-like buildings, solitary amidst fields of tall grasses, set apart from the masses but silently bursting with their own hidden throngs of society's problematic and unwanted, or their ghosts, bottled up and cavorting in conditions of squalor. It may bring to mind a ubiquitous notion of untamable chaos or a fearful notion of that which is beyond control. Perhaps, we might add an accepted meaning to this mix by deliberately associating "bedlam" with responses to and experiences of chaos, calamity, and carcerality. Perhaps, we might contextualize the term in relation to trauma, as a bodily response to trauma, or trauma's motion, bodies in states of anarchy and resistance, language itself acting in resistance to insistences of coherence or colonial insistences of meaning. Therí Alyce Pickens, whose work takes up the relationship between Blackness and madness, states that "Mad carries a lexical range that includes (in)sanity, cognitive disability, anger, and . . . excess (usually synonymous with too or really), but also acknowledges that "in modern parlance, it is used pejoratively and remains rather vague" (Pickens 2019, p. 4). Like Mad, the term "bedlam" carries with it a "lexical range" of meaning, sometimes clashing meanings, historically and culturally, and in contemporary vernaculars, it also remains vague. It is through that vagueness and the tensions around meaning, produced

under white cis-heteropatriarchal colonial power, that new cultural and literary meaning can be made around "bedlam". Perhaps, we are coming to know "bedlam" in an old and new way, as the riot of language against the violences and harms produced by sanist cis-heteropatriarchy, white supremacy, and coloniality. In this paper, I will argue that Bedlam's" linguistic riot can be re-read as a body of subversive reading and writing practices that encourage us to retool new meaning into our understandings of the relationship between trauma and madness as they are produced by and under cis-heteropatriarchal authority and the colonial transit of supremacist violences. In proposing a literary theory of "bedlam", situating it within scholarly conversations in Mad and feminist studies, I submit my own effort, as a cis white lesbian Mad-identifying woman, to ongoing scholarly and cultural efforts to subvert meaning around madness, (in)sanity, psychiatric diagnosis, and "mental illness", as they are read in bodies and bodies of work today.

Because of its associative links with madness and carceral institutionalization, "bedlam", as a literary concept, is a site of meaning that invites further scholarly deliberation. Where word origins are concerned, "bedlam" is thought to have appeared first in the form of the Old English "betleem", a reference to Bethlehem of Judea (Bedlam n.d.). This word origin, often forgotten or unconsidered, is important to an effort to unpack contemporary understandings of and uses for the term in relation to oppressed peoples and their exoduses out of systems of enslavement, violence, and oppression, both historically and within present-day webs of meaning around mental health and Madness. The word, given one of its place origins and the story of exodus associated with that place origin, is lexically, representationally, and historically linked with movements of bodies and specifically with a collective and concerted gathering and movement out of oppression by an oppressed people.

In the 13th century, with the rise of Catholicism, Bethlehem Hospital, a charitable hospital run by the priory of St. Mary of Bethlehem, began to take in those suffering from anguish of the mind and spirit (From Bethlehem to Bedlam–England's First Mental Institution). By the start of the 15th century, it was known to house people who were then referred to as "lunatics"; Bethlehem was shortened to Bethlem by London commoners and was pronounced "bedlam" (From Bethlehem to Bedlam—England's First Mental Institution n.d.). Unsurprisingly, a sweepingly disproportionate number of those inhabiting Bethlem were members of oppressed communities: the poor and marginalized on whom the church extended its moral duty in the form of shelter accompanied by conditions of isolation and punishment (From Bethlehem to Bedlam—England's First Mental Institution n.d.). In "Bardolotry in Bedlam: Shakespeare and Early Psychiatry", Benjamin Reiss describes the Elizabethan (Shakespeare's) age as one "in which insanity was generally viewed as a divine visitation, and in which only one asylum, Bethlem Royal Hospital (popularly known as Bedlam) . . . was in existence" (86). Shakespeare's constructions of madness, he argues, were used to establish a model of psychiatric authority (Reiss 2008, p. 93). He refers to Paul Starr's work in *The Social Transformation of American Medicine* to describe the ways in which asylum superintendents two centuries later, in the 1800s, drew on Shakespeare's work and pop culture appeal to attain authority (through their closeness to pop culture) over cultural meaning (Reiss 2008). In the 17th and 18th centuries, the term "bedlam" shifted away from a divine concept, becoming associated with a place name for lunatic asylums and madhouses, as well as a figurative term for manifestations and outbursts of lunacy and madness that were judged as moral failures.

Situations or scenes of mayhem, disorder, chaos, or confusion in Shakespeare's work were reinterpreted, according to Reiss, through the making of psychiatric authority during "the rationalist age" (2008, p. 102). This shift's pattern is traceable by its early modern appearance in, for instance, Dekker and Middleton's early-17th-century play, *The Roaring Girl*, in which Sir Alexander muses, "Bedlam cures not more madmen in a year/Than one of the counters does; men pay more dear/There for their wit than anywhere" (Dekker and Middleton [1611] 2011, p. 60). This passage verifies the notoriety of Bethlem Royal Hospital to playwrights and Elizabethan audiences, but it is also reflective of the period prior to the

age of rationality, which, Reiss argues, Shakespeare's work helped to anticipate and set in motion. "Bedlam" can be read in the passage as both object and subject (ivity): it is both a cure to madness, and therefore constructs Mad subjectivity, and a location where madness is gathered and objectified. Bedlam, in the reference, is not one particular hospital but a trope—a universal type of hospital where men "pay" for "their wit."

Foucault's tracing of madness in *Madness and Civilization* helps to contextualize this 17th-century use of "bedlam" by citing what precedes it. He posits that the "formulas of exclusion" applied to those with leprosy in the mid-15th century never disappeared; instead, they were "repeated, strangely similar two or three centuries later;" such formulas, he identifies in the literary, or imaginary, "landscapes of the Renaissance", which he argues is manifested through literary references to the "Ship of Fools, a strange 'drunken boat' that glides along the calm rivers of the Rhineland and the Flemish canals" (Foucault 1988, p. 7). Foucault reaches back for traces of these "ships of fools" in stories and literary lore dating back as early as 1399, tracing an imagined pattern of the expulsion of madmen driven out of the town limits; however, he is also careful to warn us that "it is not easy to discover the exact meaning of this custom", concluding that "madmen were thus not invariably expelled" (p. 9). By locating madness in the concept of the "ship of fools", Foucault draws our attention to how the "haunted imagination" invoked this Renaissance notion of "cargoes of madmen" in "pilgrimage boats" but still co-existed with the warehousing of madmen, in a paradoxical trajectory of boundedness (imprisonment) and unboundedness (passage on the open seas) (Foucault 1988, p. 9). La Marr Jurelle Bruce deepens the development of Foucault's elaborations on the "ship of fools" by theorizing madness as location and attending to connections between the "ship of fools" and the slave ship, in "Mad is a Place", the first chapter of *How To Go Mad Without Losing Your Mind: Madness and Black Radical Creativity* (Bruce 2021, p. 1). Bruce brings Foucault's theorization of madness in *Madness in Civilization* into conversation with Hortense Spillers theorization of the captivity and removal of African persons from indigenous lands and cultures into the nowhere-ness of the Atlantic in "Mama's Baby, Papa's Maybe: An American Grammar Book." This bringing together he refers to as "an unmappable coordinate where a ship of fools crosses a ship of slaves" (Bruce 2021, p. 1). Both ships, he argues, "defy positivist history: the ship of fools because it was likely unreal; the slave ship because it is so devastatingly real that it confounds comprehension, resists documentation, and spawns ongoing effects that belie the purported *pastness* of history" (Bruce 2021, p. 3). Bruce locates the echoes and evocations of what he names the "mad diaspora" of the Middle Passage, with the destabilization of homeland into the unbelonging of "restlessness and rootlessness", in Foucault's account of the "fruitless expanse" of the "ship of fools" (p. 3).

"Bedlam", while not the same as madness, theoretically, is linguistically and historically connected to the "ship of fools", in that it has primarily existed ambiguously and conceptually, more unreal than real, as a place within the metaphoric that has conceptual implications that transpire to produce effects on its historical and sensory locatability. I propose that "bedlam", in its relation to madness and the rich and telling imaginary of the "ship of fools", can be literarily theorized in relation to these two distinct but theoretically converging ships, what Bruce refers to as "floating signposts" (Bruce 2021, p. 1). "Bedlam" functions similarly to the "ship of fools" of madness: both are located as places, but in two senses: minorly, as historical realities of which there are limited documented accounts, and majorly, as unrealities that have existed primarily in the imagination in ways that have had effects on Mad materialities, on bodies in time and space. "Bedlam", then, I would argue, is a conceptual riff on the conundrum of the "ship of fools" and the conundrum of madness, and much of its potency is reliant on the unique ways it represents the tensions between historical unreality and reality, placement and displacement, locatability and unlocatability, materiality and immateriality. "Bedlam" offers us the conundrum of madness to be or not to be one thing or another, the conundrum of being bound and of resisting boundedness, of being both exiled and detained, claimed and denied, in transit and stationed, cast off into wildernesses and locked down, in materiality and meaning.

"Bedlam", as a concept, is like the "ship of fools": it is what Bruce has termed an "icon of abject madness" (Bruce 2021, p. 5). Its meaning has assumed a strange transit, an allegorical cargo moving across centuries, crossing the Atlantic, from the United Kingdom to the United States on board the primarily allegorical "ship of fools"; it has been entangled with the enslavement of African persons and the elimination of Indigenous persons under coloniality and white supremacy and slavery, but it should not be seen as analogous to that which it is entangled with, historically and in meaning. Although "both ships were imagined to haul inferior, unreasonable, beings who were metaphysically adrift amid the rising tide of Reason . . . a positivist secularist, Enlightenment-rooted episteme purported to uphold objective 'truth' while mapping and mastering the world", Bruce is careful not to draw a simple analogy between the two but, rather, to explain how "female people, indigenous people, colonized people, neurodivergent people, and black people have been violently excluded from the edifice of Enlightenment reason" (Bruce 2021, p. 4). Thus, there are different axes of analysis through which we should re-construct "bedlam" as a theory of madness and a literary theory in which "bedlam" is read and theorized through critical attention to the ways in which madness is mapped onto and practiced by bodies marginalized under coloniality, white supremacy, and cis-heteropatriarchy.

The axis of analysis used in the reading of "bedlam", or the practice of reading for "bedlam", should take into account the specific ways in which madness is represented in and projected onto different bodies and how identity matters to Mad representations and meanings when they are read in texts. By studying "bedlam" in the writings produced by those Bruce identifies as having been "violently excluded" from Enlightenment Reason (female people, indigenous people, colonized people, neurodivergent people, and Black people), "Making bedlam", in addition to being a critical reading practice that takes identity into account in considerations of representations of madness in the works of those Enlightenment Reason has banished or put into a (com)motion of unbelonging, is also the act of writing and reading in ways that deliberately challenge certain, and sometimes multiple, forms of violent exclusions. The axes of analysis, in making "bedlam", refer to (1) what we study in writers whose works make "bedlam" or bring our attention to "abject madness"; (2) how we articulate the politics of our purpose in studying "bedlam"; (3) the forms of critical reading practices and analyses that we conduct that take into account constructions of madness that challenge the sense of place-ness or place-ability; and (4) the ways in which we consider how madness is written into meaning around bodies existing under cis-heteropatriarchy, white supremacy, and coloniality and how slavery produced under coloniality also produced meaning around madness.

There is no one, uniform form of "bedlam", but "bedlam" is a landing and unland-ing of unReason—the use of "bedlam", and how it is read or written, will depend, then, on analytic practices that consider the subject to which the concept is being applied in all of its identificational, cultural, and historical complexity. Identifying, or reading, "bedlam" in a text needs to account for subjectivity: the writer and writing situation as subject and the subject matter, as well as the lines of Reason it crosses, rejects, or defies. Although there may be few literal locations of "bedlam", as in hospitals taking that name, during the past four centuries, if we shift our meaning and focus toward the figurative "bedlam" and consider "bedlam" as the transit of meaning and ideas, then it is fair to say that "bedlam" (as a concept) did, indeed, cross the trans-Atlantic. European madhouses, housing so-called "Bedlamites" (Bedlamite n.d.), moved to North America during this "turbulent movement", as Bruce calls Modernity (Bruce 2021, p. 5), in the form of late-19th- and early-20th-century "lunatic" asylums and asylums for the "insane". The rhetorical replacement of "bedlam" with "lunatic asylum", and the literalizing of a concept in and out of materiality into a structure produced under a field in the process of gaining cultural capital and authority (psychiatry), did not undo the literary ghost of "bedlam", which haunts texts on madness, exiled from meaning, from being claimed. But Mad scholars, and readers and writers who care about Mad liberation, can claim "bedlam" as their own and shape it through their analyses, recognizing its role in the construction of madness over centuries, shaped

by colonial violence, displacement, and enslavement violence and the violence of the cis-heteropatriarchy.

These developments in the cultural and historio-linguistic trajectory of the term known today as "bedlam" are important because they position Mad-identifying and other social justice-concerned activists and scholars today to take back the term, to reconfigure and refashion its meaning, or to assign new meaning to it to deepen understandings of trauma and resilience, as well as to reimagine consciousness and care. Situating "bedlam" in scholarly conversations on madness is important because such scholarship, in Mad studies, is importantly connected to public discourses and public-reaching politics. "Bedlam's" recuperative potential is part of ongoing language work being conducted within the field of Mad studies. Margaret Price, in "Defining Mental Disability", discusses the need for a deployment of language that is inclusive and invites coalition. Price notes that "Mad is a term generally used in non-U.S. contexts, and has a long history of positive and person-centered discourses" (Price 2013, p. 298). An important aspect of the work of expanding the positive use of Mad and more positive discourses around madness within the United States is the further development of a vocabulary around madness, through the study of language that already exists and through the creation of new uses for old terms. In other words, part of making "making bedlam" involves remaking the meaning of and creating new associations with the term "Bedlam" and incorporating those new meanings and associations into contemporary person-centered literary and non-literary discourses.

Poets and writers have already begun this work, whether or not they have considered themselves Mad or allies and advocates of Mad people. My hope is that theorizing "bedlam" offers a theoretical justification, rooted in Mad resilience in the face of oppression, for reading and writing practices that are trauma-informed. As I am not primarily a trauma studies scholar and my main purpose is in theorizing "bedlam" as a Mad feminist literary tool that can be used for producing new meaning around an old concept, I want to make it clear that I am using this term, "trauma-informed", in a quite literal, colloquial way, rather than a deeply theoretical way. Regarding my use of trauma-informed, as part of a literary practice of "making bedlam", I simply mean a work written by a writer who writes from a place of knowledge about the traumas endured by individuals and communities. Anne Sexton was certainly informed as to institutional trauma under patriarchy, as a woman and as someone who experienced institutionalization, and she is a knower on the subject of institutional trauma and gender-based patriarchal trauma, and so, I consider her and her writing to be trauma-informed on those subjects. Toni Morrison, though writing fiction, was informed about the trauma of racism through her own lived experience and her deep knowledge and understanding of the collective, lived experiences of racial violence of others. She is a knower, and her work is trauma-informed: it is informed by historical violence and the personal experience of living as a Black woman in a country shaped and governed by the systemic racial and colonial violence and oppression that was hundreds of years old.

I cannot say what contributions, ultimately, this essay will make to the field of trauma studies, but I hope that I, as a psychiatric survivor and a cis-gender Mad lesbian woman under patriarchy, in writing this, will contribute to trauma studies through my efforts to theorize "bedlam" as a trauma-informed Mad feminist theory. Moreover, I consider the works of Mad people and people who are institutionalized, and the works of Black women and men, and the works of Indigenous people, and the works of trans and cis women under patriarchy, and the works of LGBTQIA2S+ people under heteronormativity to be inherently trauma-informed, as our bodies, traumatized within systems that violate and socially excommunicate us, know and keep the score. Bessel van der Kolk's seminal work on the healing of trauma reminds us that "traumatic experiences do leave traces, whether on a large scale (on our histories and cultures) or close to home, on our families . . . secrets being imperceptibly passed down through generations . . . leav[ing] traces on our minds and emotions" (2014, p. 1). In "making bedlam", through reading, writing, and critical analysis, the knower, or writer in this case, is the epistemic guide to how trauma is presented, but

reading, as a dialogic practice, invites the reader to participate in the meaning making around how trauma is conveyed in relation to madness. In reading and writing madness through the practices of "making bedlam", there should be an inherent recognition of the person with lived experiences of systemic oppression or violence as a source of information on trauma, participating in the listening required if we are truly, as van der Kolk predicts, "on the verge of becoming a trauma-conscious society" (Van der Kolk 2014, p. 349). The literary method of "making bedlam", a method carried out through language practices, character development, and plot trajectories, utilizes madness as a theme, historical referent, and/or trope to claim the narrative authority to combat unjust social systems or forms of authority. These literary practices bring attention to the problematizing of notions of order, soundness, and Reason by situating madness in analyzable relation to other forms of marginalization. Where systemic violence occurs, the body and the text become a possible site for the mayhem (bedlam-style chaos) of resistance to ensue. I urge Mad and other Mad-affirming scholars to "make bedlam" by writing and reading madness as a site of contested meanings that resists authority over meaning. "Bedlam's" etymological and linguistic origins support the reclaiming of the term by those who are trauma-informed, those who have suffered under the sanist social and carceral mental healthcare systems as we know them today and know them through Mad collective historical memory.

Feminist theorists of color have provided expert witness to the theorizing from collective trauma against the denial of marginalized subjectivities and have offered guidance in understanding how theory and analyses can come together to retool meaning. In theorizing Chicana/mestiza experience, Gloria Anzaldúa describes her multidisciplinary, multilingual, and multiethnic approach to theorizing struggles for representation, writing, "I constantly shift positions—which means taking into account ideological remolinos (whirlwinds), cultural dissonance, and the convergence of competing worlds" (Anzaldúa 2015, p. 3). Anzaldúa's contributions to theories of identity as a Chicana cultural, queer, and feminist theorist are also contributions to Mad theory as they speak to the ways in which the self, or identity, and social change are connected in the writer and the writer's work as an agent for shifts in wider-spread social consciousness. The awareness of one's position in relation to power is part of the work of "making bedlam", and the study of "making bedlam" is part of a broader Mad praxis that aims to affect change in the social and mental healthcare system's exercise of narrative, cognitive, bodily, and social authority today. In the act of "making bedlam", reading, writing, and analyzing madness in a way that recognizes tensions in meaning, madness can act as an optic for the examination of power, violence, and injustice, while also being a method for narrative and social justice agency. Staking an interpretive claim can be its own form of linguistic or literary riot whose implications extend beyond the scope of literary analysis.

Black feminist theorist bell hooks theorizes language as desire, writing, "like desire, language disrupts, refuses to be contained within boundaries"—there is a feminist form of "bedlam" theorized in hooks's words that demonstrates how language is reclaimed by those it has been used against (1994, p. 167). Referring to the ways that native languages and bodies have been stolen under colonial white supremacy and the ways that the oppressors' language and its accepted meanings perform a continued trauma on Black and Indigenous people of color, hooks writes that the intimate speech created by Black people, in addition to allowing for the resistance of white supremacy, also "forges a space for alternative cultural production and alternative epistemologies—different ways of thinking and knowing that were crucial to creating a counter-hegemonic worldview" (hooks 1994, p. 171). Hooks so importantly points to the close relationship between language and cultural production: that changing the meanings and uses of words or subverting oppressive language and language practices and reclaiming linguistic and narrative authority are connected to creative liberatory cultural production through epistemologies. Taking "bedlam" back and newly defining it is one way of performing a trauma-informed Mad feminist theoretical practice aimed at subverting psychiatric and sanist authority. The development of a liberatory interpretive, or reading, practice—which makes claims about reading and writing

practices—has implications outside of the literary world because bodies of text do not exist independently of our intertextual material bodies and embodiments. Texts, like our divergent intertextual bodies, work for liberation and organize and riot under oppression. The engagements with old and new meaning around madness, through the concept of "bedlam", are acts of scholarly riot-making that suggest that by studying the ways writers "make bedlam", by speaking against and problematizing and exposing oppression, literary scholars are participating in acts of "bedlam-making" that contribute to a Mad feminist literary theory, an intersectional Mad literary theory that aims to study madness for the greater goal of the liberation of Mad people and all people.

## 2. Ringing the Bells: Bedlam as Anti-Affirmation and Trauma Response

It seems appropriate to demonstrate an application of what one practice of "making bedlam", or dismantling the order of Reason by those whose subjectivities are denied, might look like by turning to two writers who have theorized madness and made "bedlam", in different ways and within different genres. I will first turn to the place in which I first encountered the word "bedlam", after my own institutionalization: the work of a woman under patriarchy who was deemed Mad enough for institutionalization and who wrote about the concept of "bedlam" in response to it. This analysis will address patriarchal authority over the female-bodied white subject in particular. Poet Anne Sexton, associated in the popular literary imagination with madness and known widely in literary circles for her hospitalizations and suicide, configured and explored old and new meaning around the concept of "bedlam" when she named her first collection of poetry, *To Bedlam and Part-Way Back*, published in 1960 (Sexton [1960] 1999, p. 3). Sexton's poems in this collection touch upon the themes of hospitalization, madness, authority, and trauma—reminding us that they go hand in hand. The collection also reminds us that patriarchal power's operation through the mental hospital is an operation that produces trauma. Trauma, while not always obviously or overtly named, hits our readerly gut in Sexton's cutting and curt unsentimental anti-affirmations. Some of the bluntest of her anti-affirmations line up their blows in the first poem in the collection, titled "You, Doctor Martin." Sexton's contrary and precocious title invites her readers to join with her in a chorus of command: the calling upon of authority. Together, by virtue of reading the poem, readers are thrust into a confrontation with the system and find themselves calling upon the doctor by name. By placing the "you" before "Doctor Martin", Sexton implicates every reader by placing them in the role of the doctor, the "you." At the same time that she offers them the idea of themselves as the doctor and the system, she simultaneously thrusts them into an act of defiance against both—the readerly echoing act of taking charge in a hospital environment, conveyed through the "you", is an act of opposition against the power structure of the environment itself. Her title positions the reader in two identities at once, placing the reader both in and out of the "you", both in the position of patient and in the position of doctor, thereby destabilizing the power division between the two upon which the psychiatric system relies, as well as the division of Reason, on which that power differential relies. Sexton disrupts the separation of doctor and patient by placing the "you" beside the "doctor", disallowing the sameness in power and relating that is produced when the reader–speaker refers only to "doctor." Sexton's disruption of the separate subjectivities between doctor and patient enacts a poetic, textual response to and disruption of what Michel Foucault, in *Psychiatric Power*, calls "the power of sovereignty", which is described as "a power relationship that links sovereign and subject according to a couple of asymmetrical relationships: a levy or deduction on one side, and expenditure on the other", reminding us, ultimately, that "behind the relationship of sovereignty, and which sustains it and ensures that it holds . . . is violence . . . is war" (Foucault 2003, pp. 42–43). Her disruption directs our attention to the power, the power referred to in Foucault's definition of a violent sovereignty within psychiatric power, while also dismantling it.

Luce Irigaray addresses power and feminist disruptions of the patriarchal universal "you" in a way that helps explain the importance of Sexton's command. She writes, "if we



speak to each other as men have spoken for centuries, as they taught us to speak, we will fail each other. Again . . . Words will pass through our bodies, above our heads, disappear, make us disappear" (Irigaray 1980, p. 69). Anne Sexton's, "You, Dr. Martin" disrupts the traditional patriarchal relationship order between doctor and patient through the ambiguous "you", which both implicates the "you" that is the reader and the "you" that is patriarchal authority, collapsing the separation of each kind of "you" and the oneness of the "you" and, in doing so, carrying out Irigaray's psychoanalytic feminist proposition: "let's reappropriate our mouth and try to speak" (Irigaray 1980, p. 71). Sexton's reappropriation is complicated because she is speaking as paternalistic physicians have spoken across centuries, in one sense, but she is also speaking as those under that authority who have not been allowed to speak, through the invocation of the "you". Undoubtedly, "You, Ms. Sexton", would be the expected command on the part of the psychiatrist in a mid-century mental hospital. But "You, Doctor Martin" is a role reversal, which both addresses the trauma of power in a mental hospital upon those considered Mad and reclaims the power of naming trauma through the disruption of its discourse, which brings attention to the trauma itself while not condoning it. In adopting a tone that can be seen as gender-defiant or patient–doctor role-defiant, Sexton's poetic narrator and we, as readers, can shirk, disrespect, and disobey patriarchal medical authority while also naming authority and holding it accountable, by name. But the lexical ambiguity, which makes "You, (as) Dr. Martin" a possible interpretation for the meaning of the title, speaks to the complexity of the reader's relationship with power. If the reader, then, reads the entire poem as "you", the doctor, they are implicated in the system of power to which it bears witness and critiques. If the reader assumes the role of the narrator, or speaker, then they can talk back to power. The point is that Sexton's poem encourages the reader to do both.

In this way, the poem is a form of testimony, contributing to a social history of disability through the artistic representation of disability, an exceptional contribution given the fact that so often the marginalization of disabled people has led to extreme forms of social and intellectual exclusion (Borsay 2002, p. 101). Sexton's poem provides testimony to the material and immaterial conditions of exclusion when, in her first stanza, she names "Bedlam", or the hospital, as an "antiseptic tunnel/where the moving dead still talk/of pushing their bones against the thrust/of cure" (Sexton [1960] 1999, p. 3). "Bedlam", here articulated, is a place from which one cannot ever entirely return. The place of "cure" is the place where the wound of trauma is inflicted on the body, on the psyche. Bedlamites, or patients in this context, are insubordinate spectral figures: mid-century ghouls coming up against the force of treatment. Sexton's characterization of hospital patients as the living dead reaches, or points to, the coloniality of the biomedical model. Jasbir Puar, in "Prognosis Time: Towards a Geopolitics of Affect, Debility and Capacity", refers to a "bio-necropolitical collaboration" that raises questions around debility and capacity in relation to Mad people's bodies under the violences of colonization, ones which, "confound attempts to fold easily into and out of distinctions between living and dying to reflect shifting, capacious, porous and contradictory parameters of bio and necro politics" (Puar 2009, p. 164). Though Anne Sexton's positionality is one of whiteness and cis female-bodied-ness, her words speak to a larger body of people affected by institutionalization across the centuries and particularly the institutionalized of the 1950s and 1960s, who were affected by the institutionalizations of the living dead in the decades before them, in which segregation was playing out in the belief that "African Americans lacked mental and physical capacity to handle contemporary and civilized life" (Nielsen 2012, p. 122). Sexton's whiteness and economic privilege inevitably affected her institutional experience and shaped her ability to speak to her experiences of pain and trauma in writing that was validated through a literary scene that published her work, but her observations still speak to systemic violences beyond her experience, ones that created and shaped institutional violence against marginalized bodies, ones that raise issues around personhood and who counts as alive and dead under colonization.

Her words "make bedlam" in that they speak to the crossing of lines between life and death, opening space for making meaning that is historically connected but moves the reader in and outside of Reason. Her poem participates in the work of naming that was part of the efforts that disability scholars were making after World War II to challenge "ableist ideologies that viewed people with disabilities as inherently undesirable and deficient", as part of an evolving "ideology and language of rights, discrimination, and citizenship" (Nielsen 2012, p. 155). Questions of the capacity to comprehend and exercise autonomy over one's body are a fundamental aspect of colonial carceral care systems, in which forced cure-ation, or treatment, is hinged upon the authority of settler medical Reason and the exercise of that Reason (read: opinions) over the body, often the marginalized body under its control. That force is carried out through the medico-legal determination of incapacity: that was the case when Sexton wrote *To Bedlam and Part-Way Back* and that is the case today. As such, the "moving dead"—who include a range of marked, marginalized Mad bodies that still aim to have their voices heard—are positioned against the violence of cure and its deathly hallows. Stillness and capitulation are not the responses that Sexton describes when she refers to the living dead. Instead, the dead move; they respond to trauma. Cathy Caruth theorizes what Sexton puts into poetics when she writes, "the story of trauma, then, as the narrative of a belated experience, far from telling of an escape from reality—the escape from a death, or from its referential force—rather attests to its endless impact on life" (1996, p. 7). Sexton's dead, of a mid-20th-century institutional "Bedlam", are not a silent, unmoving dead; they are a trauma-responding collective of the dead, whose movements "make bedlam" or "attest" to institutional trauma's "endless impact on life" (Caruth 1996, p. 7). If the many shapes of the marginalized, institutionalized dead are moving and responding to the trauma of an intended deathly violence in Sexton's poem, their liveliness is a rejection of the regime of Reason and its binaries, which seek to control the terms of life and death, capacity and incapacity, humanity and inhumanity.

The systems that situate the institutional setting that Sexton's poem describes are larger and more complex than Sexton, and questions of capacity affect marginalized and multiply marginalized bodies uniquely. The colonial origins of this capacity-based justification of violence dates back to the British colonies, in which mental health law "existed in tandem with 'incapacity' provisions within all common law (family, civil, criminal)", in which "the British developed a normative framework for the legal disenfranchisement of a variety of people, including 'idiots', 'insane' . . . and 'criminal tribes'" (Davar 2015, p. 219). The denial of capacity is a mechanism of cis-heteropatriarchal settler colonialism: a weapon in the machinations of institutionalization. As a development of settler colonialism, the care system referred to in Sexton's collection "directly involves contests over self-determination." Such contestation is apparent in Sexton's poem, which declares the institution a place of the talking dead, dreaming of knives "for cutting your throat" and counting "this row and that row of moccasins/waiting on the silent shelf" (Sexton [1960] 1999, p. 4). The "broader patterns of colonialism" (Burch 2021, p. 4) that Susan Burch identifies in *Committed: Remembering Native Kinship In and Beyond Institutions* are evident in the institution that Sexton describes, the authority that presides over it, and the "Bedlam" within it, those bodies of "the foxy children who fall like floods of life in frost" (Sexton [1960] 1999, p. 4). Sexton speaks to an omniscient arm of colonization, declaring: "Your business is people, you call at the madhouse, an oracular eye in our nest" (p. 4). Calling upon, or out, the lead authority presiding over the bodies that animate the actions of "Bedlam", in the movement of that forced "contest" over self-determination, Sexton's poem theorizes a "Bedlam" which practices agency within the movement of contestation with settler colonialism. Later, in her poem "Unknown Girl in the Maternity Ward", Sexton takes us into another sector of the ward, where she speaks, not to a doctor but to a child that must be given up and handed over: "You sense the way we belong", she writes, "but this is an institution bed. You will not know me very long" (p. 24). Here, inner conflict and numbness do battle in the confines of a space in which choice is not choice, and sovereignty over one's person is denied—the space is reflective of the larger practices of violence committed under and by settler colonialism.

Sexton's poem comments on the temporal nature of institutionalization, and Mad bodies that are both detailed and in motion, put in close proximity but simultaneously kept apart, forced together but severed, in flux between transport and delivery.

"And this is the way they ring the bells in Bedlam", Sexton begins in the latter segment of her collection: "we are the circle of the crazy ladies/who sit in the lounge of the mental house/and smile at the smiling woman/who passes us each a bell" (p. 28). The bells in Sexton's "Bedlam", and the smiling of the bell ringers, stand in stark contrast to the scenes of humiliation and coercion and the tone of resentment that Sexton conveys. The outward beauty of the sound produced by the bells, the note of E flat, contrasts with the internal reality of the ringing of the bells. Sexton's poem does not leave us with the sound but with the inner bitterness that "we are no better for it" and a reminder of the coercion under which the bells are ringing, for they only ring because, as the last line of the poem reminds the reader, "they tell you to go. And you do" (Sexton [1960] 1999, p. 29). The contrast between the associations of bells and pleasant sounds with the forced ringing of the bells by the bodies under institutional authority is a pattern carried out in the poem that follows, titled "Lullaby", in which the narrator is in "the TV parlor in the best ward at Bedlam" (Sexton [1960] 1999, p. 29) waiting for a sleeping pill to be delivered. The lullaby, associated with the innocence of childhood and the safety of home, is conveyed in the dystopic and warped keys of the institution, which offers comfortlessness to its infantilized residents but which lulls them, not through a lullaby but by "a splendid pearl" which Sexton writes, "floats me out of myself,/my stung skin as alien/as a loose bolt of cloth" (p. 29). She then, in her sleeping state, drifts out of herself, entering into displacement, and seeks to leave the company of the other prisoners around her: "let the others moan in secret" (p. 29), she declares. Though the poem does not offer a clear explication of whether the taking of the pill is an actual choice or a choice produced by institutional coercion, Sexton's somber tone indicates that the pill is a mode of escape from the moaning pain, held in secret in the bodies of the institutionalized. In these poems, contestation of selfhood and injurious power do battle in a way that reveals madness's production under settler colonial violence. Elaine Scarry's insights into what she calls the "transformation of body into voice", are apropos in that they help to situate Sexton's poems in *To Bedlam and Part-Way Back* within the context of what she calls "the structure of torture", in which "the translation of pain into power is ultimately a transformation of body into voice" (Scarry 1985, p. 45). "The prisoner", Scarry writes, "experiences his own body and voice as opposites", so that no matter the setting of or the reason for the suffering caused by the source of torture, "the person in great pain experiences his own body as the agent of his agony" (p. 47). Scarry asserts that there is a "sense of self-betrayal in pain, objectified in forced confession" (p. 47)—and, while referring to the structures of torture in war, specifically, her philosophy of pain offers insights into the scenes of "Bedlam" that Sexton's poems depict. The narrator's description of the act of taking the sleeping pill, of being floated out of herself, her "stung skin as alien as a loose bolt of cloth", suggests a separation of the body from consciousness or of the body and consciousness from the institution and its violences, which sting the body and from which the body seeks the solace of escape. "Bedlam", as an institutional location of colonial violence, is a place of pain and trauma infliction, a place in which one's sense of self is betrayed as much as one's bodily autonomy, a place from which one's consciousness can never fully leave; but, as Sexton's poems indicate, it is also a place rife with the chaos and contestations of resistance.

The "bedlam" *made* within that place, that which makes it a place of—and not just called—"bedlam" is made by the resisting bodies and spirits of the institutionalized. Bedlam as a form of resistance can be made in endless ways, but it is always resistance against systemic, and often carceral, violence. It will be helpful for us to move away from thinking of "bedlam" as a location in order to move toward seeing and making "bedlam", or seeing and making resistance against colonial, white supremacist, and sanist cis-heteropatriarchal carceral power and violence, especially within the psychiatric institutional context. Whether in the unheard echoes and moans or in the articulations of the displacement from the self

that take place as a result of institutional violence or in literary representations of institutional violence or in a vastly divergent manifestation, the notion of "making bedlam" is not a new one; it is one that is apparent in the literary and autoethnographic discourses that center on institutionalization, colonial violence, white supremacist violence, and sanist cis-heteropatriarchal violence. All of these aforementioned violences exert their supremacist powers over both the body and the marginalized representational self, or the marginalized selfhood of a people, both of which are part of the act of claiming and exercising authority over the bodies that systemic violence relies upon.

## 3. Making Bedlam: Trauma-Informed Literary Analysis as Social Justice Literary Practice

In the way that Susan Burch postulates that "attention to settler colonialism and Indigenous self-determination redirects historical interpretation", resisting "a view of time and place dictated by a singular institution ['s material opening or closing]" (Burch 2021, p. 4), it is important to redirect the historical interpretations of "bedlam", as a carceral conveyor of settler colonial, white supremacist, and cis-heteropatriarchal violence, and to pay attention to the self-determination of Black and Indigenous People of Color, LGBTQ+ people, women, and Mad people whose bodies and lives, material and symbolic, have been affected by its violence. Sexton's constructions of "bedlam" do not do what Mad studies scholar Erick Fabris warns us against: they do not "normalise feelings and ideas and make them consistent with common sense" in order to support "the experiences and thought already inscribed" (Fabris 2016, p. 99). Instead, the constructions embrace the non-normative through the conflicts and contrasts and contestations they represent, which are produced under institutional conditions. They do not "[glory] in understandability" or attempt to respond obediently to "psychiatry demanding understandability from its wards" and "[making . . . demands] on survivors for readable prose" (Fabris 2016, p. 100). Sexton's poetry encourages readers to engage in Mad reading practices, ones aimed at denying the demands from the arbitrators of sanity and capacity under colonial systems of violence and the patriarchy-powered arbitrators of Reason. Her constructions of "bedlam" as a place help us to recognize "bedlam" as a method, but they do not address the vast scope of colonial violence to which "making bedlam", as a methodology of resistance, is capable of responding. For this reason, it is important to understand her work in its historical context as well as to understand "bedlam" as a complex location into which oppressions and systemic violences need to be read.

Sexton's poetry collection contributes to a United States disability history in a way that challenges psychiatric oppression and that reflects that the disability history in the United States is, indeed, "a complicated and contradictory story . . . of lands and bodies stolen . . . of rights and wrongs, of devastation and ruin . . . and of the reinvention of self" (Nielsen 2012, p. 182). Though Sexton's work speaks to systemic violence, broadly and particularly to patriarchal violence's effects on women's and Mad bodies, what Sexton's work cannot speak enough to that a feminist theory of "making bedlam" needs are the ways in which "the Euro-modern patriarch affirmed his Reason and freedom . . . by casting the black African as his ontological foil, his unreasonable and enslaved Other" (Bruce 2021, p. 5). Looking to writers whose work does not address madness and institutionalization directly but whose work deals with colonial and white supremacist and cis/hetero/patriarchal violence is an important part of demonstrating that "making bedlam" has been and can continue to be a tool for recognizing the complex ways in which colonial violence operates—its effects on Mad, neurodivergent, and "otherwise othered" bodies, often multiply marginalized bodies, within societies and institutions. These need to be understood through a complex and intersectional Mad feminist analysis in order to be a useful praxis which can function for the liberation of all bodies marginalized and harmed under patriarchy, white supremacy, and colonization. As a second application intended to demonstrate another shape that "making bedlam", as a reading and writing process, might take, I will turn to a well-known and much-written-about work that addresses white supremacist colonial violence

and that demonstrates the role that fiction plays in "making bedlam" by speaking to historical traumas through historical representations of madness as an agential literary trauma response.

In the forward to *Beloved*, Toni Morrison writes that "to invite readers (and myself) into the repellant landscape [of slavery 'formidable and pathless'] (hidden, but not completely; deliberately buried, but not forgotten) was to pitch a tent in a cemetery inhabited by highly vocal ghosts" (Morrison [1987] 2004, p. XVII). The "highly vocal ghosts" who resist oppression and violence and who riot for justice in Morrison's novel embody a quality and function of madness when it arises out of legacies of trauma caused by systemic identity-based forms of violence. Madness, in the way that Morrison uses it, is not insanity, and it is not merely a reaction to violence or a response to trauma; it is a practice of resourcefulness and agency, as well as a narrative method that encourages readers to think not only about how madness functions within literature but about how it functions and can function outside of it. This is "making bedlam": a purposeful and agential trauma-response as a literary and liberatory method. It functions both in the realms of literary theory and literary writing, in which interpreters read, name, and utilize social justice action in literary narratives and in which social justice writers work for systemic change. Morrison's introduction claims, names, and gives readers a lead into the social justice aims of her work. She makes no bones about it but, instead, directs her readers to this knowledge: that the ghosts of *Beloved* use madness—a relentless and vocal inhabitance—as a form of agency that combats racism and slavery, seeks justice, and paves new paths for change. In framing her literary agents (ghosts) this way, Morrison urges readers to think expansively about time, history, and literary agency; to consider the potential of literary madness to act as a vehicle for social change; and to recognize that what is cast under the heading of "contemporary literature" bears close ties to the histories, systems, and power dynamics that produce it, that were formed long before it, that exist outside of it, and that continue to exist. Morrison's historical fiction is trauma-informed and framed this way. In *Quiet As It's Kept: Shame, Trauma, and Race in the Novels of Toni Morrison*, J. Brooks Bouson examines the complex ways in which Morrison's fiction "focuses on inter- and intraracial violence . . . even at the cost of alienating, or even unsettling and hurting, some of her readers" (1999, p. 3). We get a sense of this in her preface, which sets the stage for understanding her own "bedlam-making", as she makes literary representations of madness a trauma-informed agential political practice. It encourages us to assume and interrogate the literary agency of madness, to consider the ways in which writers and their characters are "making bedlam", and to be deliberate in "making bedlam" as an intersectional literary social justice reading and writing practice.

Problematic parts of the Mad movement's history, in which the movement has perpetuated its own "mechanisms of exclusion and domination" have had to be rethought (Russo 2016, p. 65). This rethinking is part of a Mad feminist literary theory and praxis and part of rethinking trauma so that the resistance against oppression does not adopt uncritically its own oppressive dynamics—"making bedlam" can help us to do that if we are not unobservant and are connected to our analysis, actively (re)thinking the mechanisms of exclusion and domination as they appear in literary contexts and as they exist outside of them. "Making bedlam" is both relevant to speaking back to histories and into gaps or erasures in histories and to doing the work of making meaning around madness, trauma (historical and other), and liberatory resistance.

By helping readers to see the ways in which power operates and the intricacies and sweeping effects of its violences against bodies targeted as *other*, and by representing agents of "bedlam", Morrison's work offers a social critique of the notions of harm and trauma that can be used to reimagine and reform social systems, especially the systems of care which lay claim over madness as a subject, or the subjectivities of madness. By (re)claiming and continuing to claim madness as a literary and social justice subject, and by paying attention to the way that writers do this deliberately and unconsciously, we can uncover new understandings of individual and intergenerational trauma and trauma responses, as

well as create the necessary chaos and friction that produce liberatory change. The tools best used for Mad liberation are those which work for the liberation of all marginalized peoples. Audre Lorde's words in "The Master's Tools Will Never Dismantle the Master's House" are just as relevant and important in the context of "making bedlam" for Mad (and other social) justice as they were when she wrote and spoke in 1979: "What does it mean when the tools of a racist patriarchy are used to examine the fruits of that same patriarchy? It means that only the most narrow perimeters of change are possible and allowable" (Lorde 2007, pp. 110–11). Mad scholars, such as myself, have a responsibility to other movements to work for intersectional justice in the work we do. It is not a perfect practice, but there is no liberation for Mad people without the liberation of all people. Literatures of "bedlam", in which madness can be re-read as liberatory agency through close and careful intersectional literary analysis have the potential to join in the work that Mad scholars, such as La Marr Jurelle Bruce and others, are doing to "approach madness as an object of analysis" (2021, p. 9) and to engage in a mad methodology, which Bruce defines as "how to go mad without losing your mind", an acknowledgment of madness's mindfulness and its ways of "frustrat[ing] interpretation" and "resist[ing] intelligibility" (2021, p. 11). The "bedlam-making" of going Mad that Bruce describes can be traced in both autoethnographic poetry and historical fiction, through the subject of the Mad white woman poet under patriarchy and the subject of the Black woman writer under white supremacist patriarchy.

## 4. From Trauma Response to Mad Agency: Social Justice Transliterary Methods, from Autoethnography to Truth-Telling Fiction and Back Again

"Making bedlam" is important to developing our understanding of trauma, not just from a literary standpoint, but also from a political and social standpoint. The reading of madness, the reframing of madness as an agential trauma response, and the utilization of madness as a social justice practice can be elements of the study of the literary device of "bedlam", in which writers can strategize and organize literarily and otherwise under systems of oppression. A "bedlam method" of reading, writing, and enacting madness as practice reads and writes madness as both trauma response and a social justice tool, recognizing that it opens itself to Mad understandings of and (re)interpretations of language and meaning, as well as to ways of seeing, locating, and naming histories, particularly erased, ignored, or suppressed histories. Given this, part of "making bedlam", is attunement to trauma writ large and writ small and, in particular, to the epistemic violence of the "institutional processes and practices committed against persons or groups, such as Aboriginal peoples, that deny their worldviews, knowledge, and ways of knowing, and, consequently, efface their ways of being" (Liegghio 2013, p. 123). Where there is trauma, especially epistemic trauma caused by systems of violence, we should be looking for the agency of madness and seeking to understand how it is functioning to undermine, dismantle, speak back to, speak a way out of, or speak a way beyond violence and oppression. It is in "making bedlam" or in turning to worldviews, knowledges, and ways of knowing and being that have been denied and violated that we can challenge epistemic violence and its institutions. The challenges that take place in our literatures and in our public discourses, and discourse mayhem against structures of epistemic violence, are likely to effect change in our systems, dismantling the tyranny of certain subjectivities over others and creating space for more subjectivities to peacefully and respectfully coexist as equals, on equal grounds.

Toni Morrison's commentary on the "highly vocal ghosts" that arise in the context of slavery alerts us, as readers, to a claimed and deliberate trauma response (2004, p. XVII). Her forthrightly expressed intentions for framing the novel are made even more clear when she names a major source of inspiration for the novel: Margaret Garner. Garner, Morrison tells us, was a young mother "who, having escaped slavery, was arrested for killing one of her children (and trying to kill the others) rather than let them be returned to the owner's plantation" (2004, p. XVII). Morrison frames this historical framing by

referring to readerly reception, stating that it was "her sanity and lack of repentance" that "caught the attention of Abolitionists as well as newspapers" (2004, p. XVII). Morrison's reference to this newspaper clipping about Margaret Garner is important to a reading of *Beloved* and to interpretations of representations of madness in the novel. It is also relevant because it asserts a response to the collective trauma of epistemic violence posed by the violences of white supremacy and enslavement. But this is all part of a writer's introduction, a writer's method. Morrison further delves into her method for synthesizing the novel, revealing that her method would be to expand the "imaginative space" through the form of the novel to create "a subtext that was historically true in essence, but not strictly factual in order to relate her history to contemporary issues about freedom, responsibility, and women's 'place'" (2004, p. XVII). To do this is to "invite readers into the repellant landscape" of the "formidable and pathless" terrain of slavery; it is to bring back a ghost— to have her "enter the house" (p. XVII). This is not just an invitation into a history of the violence of enslavement; it is also an invitation into a world in which Morrison has authorial, epistemic control: a narrative world over which she claims control of the history represented and the ways in which it is represented, including the ways in which violences and collective traumas are articulated. "Intent on representing race matters in her novels", Bouson emphasizes, "Morrison repeatedly, if not obsessively, stages scenes of inter- and intraracial violence and shaming in her novels" . . . using "her fiction to aestheticize—and thus to gain narrative mastery over and artistically repair—the racial shame and trauma she describes" (Bouson 1999, p. 19). The styling of trauma, or the aestheticizing of trauma, that Bouson describes participates in what Bruce calls "psychosocial madness: acts and attributes such as insurgent blackness, slave rebellion, willful womanhood, anticolonial resistance" and in what Bruce calls "phenomenal madness" (Bruce 2021, p. 8). The distinction, according to Bruce, is that "psychosocial madness" refers to an "unruliness of will", whereas "phenomenal madness" refers to "an unruliness of mind" (2021 p. 8). Morrison practices a waywardness of will through her stylizing of trauma as a historical social justice method; her characters, or ghosts, exhibit "phenomenal madness", through the waywardness of mind and form. It is through an insistence on an unReasonable sanity, in the form of historical (in)sanity cast into literary madness, that Morrison demonstrates this.

Morrison's (re)assertion of Garner's sanity is central to understanding the representations of madness in the novel, as they seem to topple the notion of madness as a form of insanity, making it something other, something outside of the sane/insane binary, and offering it (madness) the agency of sanity while also still manifesting in ways that would evoke the trope of madness. It is not possible to ask Morrison about her use of the term "sanity" and to ask that she expound further upon this preface and the issue of madness, but it is possible to consider her preface's framing of the novel as a response to the trauma of enslavement, which forces upon the body, if we consider Morrison's paradox, a sane and unrepentant madness, a madness explored through the truth-telling and testimonial justice literary trauma responses of Morrison's imagination. Morrison is making claims on sanity and is therefore asserting Garner's capacity as a knower. In this way, she is addressing an issue of testimonial injustice that might arise in relation to contemporary views of the act of killing one's child within the context of the complex and acute historical trauma of slavery. Miranda Fricker argues that the wrong of testimonial injustice is an intrinsic harm that "can inhibit the very formation of subjectivity" and that it is also a "distinctive form of objectification" (Fricker 2007, p. 145). Madness, when pathologized through an institutional lens, sweepingly results in testimonial injustice: the testimony of the speaker is denied because the speaker is denied subjectivity and, therefore, is exposed to the violence of being denied the capacity to be a "giver of knowledge" (Fricker 2007, p. 145). Morrison's preface alludes to issues of testimonial injustice by validating Garner as a knower by asserting her sanity; in this case, sanity is the capacity to know and produce knowledge. It is not necessarily the opposite of madness, and so, madness is not necessarily what is being pushed away by the claim of sanity that Morrison makes.

The label of "madness" often carries with it stereotypes that "become the mechanisms used to diminish and deny . . . people their social legitimacy as knowers" (Liegghio 2013, p. 123). Morrison's move to name the inspiration of her novel as a knower, through her characterization of her and her actions is a way of framing her novel as a fictionalized reclamation of epistemic knowledge—and it is this "sane madness" of the trauma-speaking and mayhem-causing ghost protagonist that addresses epistemic violence by vehemently insisting upon epistemic knowledge. Simultaneously, Morrison divests herself as a writer of the burden of representing an entire subjectivity and instead writes testimonial justice on the subject of slavery by forming a subjectivity of the dead, come back to life to haunt the living; this is the trauma-informed epistemological "bedlam" of *Beloved*.

In so much as Beloved is a ghost seeking testimonial justice, insisting on her own denied subjectivity, Morrison creates within fiction the space for testimonial justice to take shape where slavery is concerned: she asserts her own subjectivity, through the novel's testimonies to its violences, including the denial of knowing, the denial of subjectivity, and objectification. But Beloved is not the only ghost and not the only maker of "Bedlam". As a literary and social justice device for "making bedlam", or mayhem, within the context of white supremacy and enslavement, exposing the mayhem of trauma that slavery imparts, Morrison's Mad ghostly literary representations of knowing and knowers go beyond and fracture the designations for knowing and knowers that white supremacy recognizes or claims (that is, its own knowing and its own practitioners). The house itself is a knower: it offers an epistemic embodiment of trauma and sane madness: "124 was spiteful. Full of a baby's venom. The women in the house knew it and so did the children" (Morrison [1987] 2004, p. 3). Sethe and Denver's house, haunted by the painful sane madness of epistemic and systemic violence, has made them, at the end of the 19th century, "its only [its last remaining] victims" (p. 3). Morrison's method gives the house itself its own capacity for knowing; it is a vengeful knower, and its knowing, its spite, its sane madness, for spite always asserts some claim on knowledge, (re)produces the trauma imposed by slavery onto the body, self, family, and community onto its inhabitants. The "bedlam" of the household, the "kettleful of chickpeas smoking in a heap on the floor" and "the lively spite" which caused its past inhabitants to flee, can be seen as an embodied trauma response of sane, or justified, madness, in which the body, under the tyranny of systemic violence, revolts.

In *Black Madness :: Mad Blackness*, Therí Alyce Pickens states that "in an ideological construct of white supremacy, Blackness is considered synonymous with madness or the prerequisite for creating madness" (Pickens 2019, p. 4). Pickens's recognition that the construction of madness is produced by systems of colonization and white supremacy, and that its construction is produced to maintain epistemic and systemic supremacy and violence, provides an epistemological touchpoint and foundation for engaging in the analysis of literary and pop culture representations of madness that are deeply embedded in the social fabric and institutional fixtures that perpetuate colonial and supremacist power and uphold its epistemic authority. Pickens importantly notes the many meanings of madness: that the term "Mad carries a lexical range that includes (in)sanity, cognitive disability, anger, and . . . excess" (p. 4). In Beloved, the sane madness of anger, represented by both the house and the soul of a baby girl, serves to name trauma, infuse it with life, and render it agential: trauma is excessively alive; it is so much alive that it is living beyond death, which is a meaning of "haunt." There is a deliberateness in the act of haunting that connects with the definitions of madness that Pickens describes, and it is the deliberateness and the knowing inherent in that deliberateness that renders the excess, anger, suffering, and struggle of madness "sane". When Morrison writes, "Who would have thought a little old baby could harbor so much rage" (p. 5), she is confronting her readers with the linguistic and ideological mayhem of double paradoxical juxtapositions: not only of "old" and "baby" but of "baby and "rage"—and, therefore, of innocence and rage. Morrison makes madness knowing by putting it in the most innocent of character embodiments, a baby. She makes a baby a foremost knower, and the knowing of that baby manifests as

the "sane madness", or the "bedlam", of excess and anger, two forms of madness that are suppressed under white supremacy precisely because they revolt against it.

House 124, a house haunted by the reverberational violence of slavery, is not a "normal house" (Morrison [1987] 2004, p. 49); it is defined by its excesses of "strong feelings" (p. 47). The house's embodiment of trauma and the transference between that and the ghost of Beloved's embodiment both speak to what arises in conversation between Morrison's protagonists, Sethe and Baby Suggs, when Baby Suggs declares, "Those white things have taken all I had or dreamed . . . and broke my heartstrings too. There is no bad luck in the world but whitefolks" (pp. 104–5). Baby Suggs's knowing manifests specifically in knowing and naming the effects of white violence on the household: "124 shut down and put up with the venom of its ghost", which leads to Morrison's commentary on trauma, when she writes that "Sethe knew the grief at 124 started when she jumped off the wagon, her new-born tied to her chest in the underwear of a whitegirl looking for Boston" (p. 105). Morrison asserts the knowing of her characters, their subjectivities demanding the authority that Garner and others affected acutely by such violence historically named through their "sane" revolt and resistances. The women who are the knowers in Beloved reach into the places historical memory cannot always reach: identifying the source of the grief that constitutes the madness produced by white supremacist and colonial violence. Morrison's act of "making bedlam" is one that requires that the reader's "understanding of the world" is "engaged—in order to be confirmed or disrupted" (Pickens 2019, p. 14). In the case of Beloved, disrupted, for disruption is a core theme of the novel itself, is accounted for in "the undecipherable language clamoring around that house" that "was the mumbling of the black and angry dead" (Morrison [1987] 2004, p. 234). In part trauma response and in part moral confrontation and in part epistemological reclamation, the novel confronts readers with the fact that "no reader is innocent" and that "no reader can be divorced from discussions of race in American letters" (Pickens 2019, p. 14). Pickens makes an important argument on the import and impact of readerly responsibility in reception when she states, "to read Blackness and madness then, to participate in such reading, requires that readers bear the responsibility of interpretation: understand that multiple interpretations are available and that their choices indicate a stance on Blackness and madness itself" (p. 14). This assertion on Pickens's part collaborates with Susan Burch's and Hannah Joyner's acknowledgement that "historians have not merely remembered; we also have misremembered, dismembered, and disremembered the past" by isolating "specific identity vectors (such as race or gender) or specific social forces (such as oppression)" (Burch and Joyner 2019, p. 65). The acknowledgement of mis- and dis-remembering by historians and within the study of history is crucial to the study of madness as it relates to systems and histories of oppression, as well as to efforts to liberate those whose marginalization has allowed madness to be weaponized against them.

"Disremember" is Morrison's word and concept, which places on readers the responsibility in reception to grapple with "deliberate effort[s] to escape from painful memories" that are "never truly dead" and that continue to "inform the present" (Burch and Joyner 2019, p. 66). What is "forced out of consciousness", as Burch and Joyner remind us about Morrison's work, still lives and performs its knowing, articulating its knowledge, upon the present (p. 67). While Burch and Joyner consider Morrison's notion of "disremembering" in relation to disability history to "envisage a theoretical framework for deaf cultural history" (p. 67), Pickens's insistence upon the acknowledgement of the ways in which Blackness and madness have been written into one another under colonial tyranny and white supremacist ideology forges a theoretical framework for Mad cultural history that must always consider how race, how Blackness, how Indigeneity, how sexuality, how gender, and how other vectors of identity are coproduced in relation to madness and how madness is produced through and within them under colonization and white supremacy. This is a crucial dimension of the work of Mad studies; it is also a crucial dimension of tracing trauma through the "bedlam" of resistance and revolt and of "making bedlam" as a social justice readerly and writerly practice. This is the reckoning that the "bedlam" of disremembering anticipates

and stirs: the reckoning of reflection upon cultural historical culpability. Morrison's novel brings us into the mayhem sprung from systemic violence:

> It was the jungle whitefolks planted in them. And it grew. It spread. In, through and after life, it spread, until it invaded the whites who had made it. Touched them every one. Changed and altered them. Made them bloody, silly, worse than even they wanted to me, so scared were they of the jungle they had made. The screaming baboon lived under their own white skin; the red gums were their own. (Morrison [1987] 2004, p. 234)

Morrison's "bedlam" insists on culpability: it is the revolt of justice, righteous excess. It is an unstoppable mayhem whose violence spreads and penetrates everything but whose chaos is called to justice by the readerly reception of the conscience, which Morrison stirs and calls to answer.

Her prose preceded Pickens's proposal that "the mad Black" be a way to "reimagine and reread" (Pickens 2019, p. 52). Collective trauma, inflicted by white supremacy and colonial tyranny, is accounted for in the "bedlam", or mayhem, of trauma, in "the voices surrounding the house, recognizable but undecipherable . . . the thoughts of the women of 124, unspeakable thoughts, unspoken" (Morrison [1987] 2004, p. 235). The accounting for of trauma does not articulate itself in the spoken but, rather, in the unspoken and unspeakable, defining race as a matter of "life and death" (Pickens, p. 24). Morrison's naming of and insistence upon the unspeakable does not "foist assemblage onto intersectionality", "reduc[ing] Black women's embodied theorizing" (Pickens 2019, p. 18). Instead, it pushes against the category of the human, which under colonial and imperial control was and is "designed to exclude Indigenous people and Blacks (Pickens, p. 76). Categories of the human, of time, and of history are all complicated by "the mumbling in places like 124" (Morrison [1987] 2004, p. 235). It is helpful to consider Pickens's assertion that "Blackness appears as the antithesis of history, its excretion, whereas whiteness stands in for progression, being in time" so that in order to consider the Black Mad subject, "we must consider that this person is meant not only to occupy space but to be consistently removed from space in order to make room for the more recognizable subject: the white able body" (Pickens, p. 29). Morrison's method holds readers to account for the chaos imposed by supremacist violence, the "bedlam" that signals the refusal of the whitewashing of history and the mumblings of systemic trauma that, when reclaimed as historical knowing, dismantle the category of the human under white supremacy and insist upon Black subjectivity. Morrison writes, in the final paragraphs of the novel:

> There is a loneliness that roams. No rocking can hold it down. It is alive, on its own . . . Everybody knew what she was called, but nobody anywhere knew her name. Disremembered and unaccounted for . . . so they forgot her . . . The rest is weather. Not the breath of the disremembered and unaccounted for, but wind in the eaves, or spring ice thawing too quickly. Just weather. Certainly no clamor for a kiss. Beloved. (pp. 323–24)

Readers are returned to the disremembered child and to a reminder of the testimonial violence and injustice—the trauma—of disremembering, its deliberate erasure, against which the ghosts of epistemic justice utilize the method of madness to account for trauma and insist upon subjectivity. Morrison's method is deliberate; as Bouson claims, it is a way of "demanding participatory reading and having both a cognitive and emotional impact on readers . . . exerting interactional pressures on readers" (Bouson 1999, p. 20). Her "bedlam" is a writerly rhetorical move which subjects the reader to the trauma of history in a way that demands that they participate in it actively.

Though pushed out of mind under the white supremacist violence of disremembering, Morrison's "bedlam" asserts Black subjectivity, its calls for historical conscience and restitution, and its knowing, which is, finally, given a name by Morrison: Beloved. The testimonial injustice of racial systemic violence, in which one is denied access "to what originally furnishes status as a knower" is not undone but it is challenged (Fricker 2007,

p. 145). The idea of a knower under white supremacy is challenged through Morrison's making of "bedlam", the work of excess and anger put toward an insistence on the subjectivities of the subjugated. A reading of Morrison's chronicling of a history of unReasonable resistances to white supremacist violence as "bedlam" is part of the work of a Mad feminist literary theory that aims to re-shape the discourses that rely on Reason, participating in a mad methodology in the way that La Marr Jurelle Bruce defines it: as a "mad ensemble of epistemological modes, political praxes, interpretive techniques, affective dispositions, existential orientations, and ways of life", historicizing and contextualizing "madness as a social construction and social relation vis-à-vis Reason" (Bruce 2021, p. 9). Morrison's work engages in a Mad methodology in the ways in which it "resists the hegemony of positivism" (Bruce 2021, p. 10) and, instead, grapples with the trauma of slavery. In *Beloved*, she does this by "examin[ing] the white supremacist ideology and essentialist discursive repertoires that defined the African American slave as the racial Other . . . " and also by "dramatiz[ing] the social and political consequences of racist thinking and practices" (Bouson 1999, p. 131). This literary work of "making bedlam", of challenging the order of white supremacist Reason and of resisting positivism is part of a larger movement of Mad liberation work.

"Making bedlam" responds to traumatizing violences performed under that system on bodies marked as Mad and on bodies marked as Mad because they are symbolically marked as Other in other identity categories which are co-constructed by madness and which co-construct madness. As a literary social justice practice, its intersectional feminist analytic study aims to produces a more trauma-informed and historically-responsible result that challenges, dismantles, and re-forms unjust social systems and institutions of violence that abuse power and inflict harm upon bodies. Language is a central part of acts of oppression and acts of liberation. "Making bedlam" is language work that deals with and strives to initiate change for bodies struggling against systemic violence across time. The language of "bedlam" is the language of the Mad and the madness of language which riots against the strictures of Reason, as "a proper noun denoting a positivist, secularist, Enlightenment-rooted episteme purported to uphold objective 'truth' while mapping and mastering the world" (Bruce 2021, p. 4).

The term "bedlam" should not be relegated to a sentimentalized relic of the past, as its potential to affect change in the lives of Mad people today depends on its reclamation and expansive reassignment of meaning. Within queer studies, Robert McRuer's critical act of "cripping" addresses embodiment while resisting the "straightening" of time and space, drawing on the work of Disability Justice scholar–activists—"crip", unlike the term "queer", he argues, has not been "domesticated" or "contained and commodified" (McRuer 2018, p. 24). Sins Invalid, a Disability Justice disabled queer of color art space and performance project formed through kinship, grounds its philosophy in intersectionality, acknowledging the intersections of "bodyminds" and creating new imaginings for "bodymind" experiences through art (Sins Invalid). Among their "Ten Principles of Disability Justice", published both in *Women's Studies Quarterly* and on their website, is the principle of "leadership of those most impacted", an insistence on leadership in endeavors related to Disability Justice that prioritizes the voices, perspectives, and leadership of those affected most by ableism, white supremacy, and heteropatriarchy under capitalism, with an emphasis on cross-movement and collective liberation (Berne et al. 2018). Word choice and word consciousness are primary fixtures in the past and future work of both the Mad movement and Mad studies because pathologizing language has been used in the systemic oppression of and in violences against disabled and Mad bodies, and Disability Justice strengthens and elevates the work being conducted around language by offering a framing grounded in the work of those whose lives have been multiply marginalized under capitalism and its production of sanist discourses and practices of harm. This paper, written by a Mad-identifying scholar, aims to contribute to such liberation efforts.

Poets, authors, and autoethnographers writing on madness provide Mad scholars and scholars interested in justice more broadly with representations of madness as agency that lay the groundwork for a "bedlam" theory and praxis, or a methodology of mayhem—a

model of working for Mad and other forms of social justice liberation. Such groundwork is important to scholarships of resistance and is connected to liberation movements. This is both a literary and a community-focused endeavor. By naming and describing states of madness (in the form of discomfort, rage, emotional distress, excess, ecstasy, or anguish), writers and readers engage in social justice literary practice, aimed at deepening our historical and collective cultural understanding of how systems of oppression produce trauma and how madness functions as a trauma response that is not passive but instead includes refusal and rebuttal and revolt. By closely considering madness as a valid subjectivity with something important to say and do, it is possible to begin to more fully understand how manifestations of madness, both on and off the page, function. Agential responses to trajectories of imperialistic order and power-based violence in literary and non-literary contexts are already doing the work of "making bedlam". Our important work as readers and writers engaged in the re-remembering of the subjectivities of madness offers us an opportunity and responsibility to make our own social justice mayhem in an effort to dismantle injustice and work for liberation.

**Funding:** This research received no external funding.

**Institutional Review Board Statement:** Not applicable.

**Informed Consent Statement:** Not applicable.

**Conflicts of Interest:** The author declares no conflict of interest.

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
