# Peer review of "Making Bedlam: Toward a Trauma-Informed Mad Feminist Literary Theory and Praxis"

_humanities, doi:10.3390/h12020024_

Round 1
Reviewer 1 Report
I find this article to be very well researched and written. While I sometimes would have wished for a bit more precision (some sentences run on, some paragraphs feel unnecessarily long, some examples could have been shortened), I overall appreciate the deep level of engagement and care that the author brings to this piece.
I do have one significant worry: Anne Sexton is white and, to my knowledge, not Indigenous. Using Puar and Burch's thinking to discuss a white, non-Indigenous author seems unethical to me. I would encourage the author to explicitly grapple with the fact that Sexton is white, in the piece itself, and then to analyze what this does to the argument of 'making Bedlam.' Some of the statements in this section, despite the ubiquitous naming of settler colonialism etc., universalize a particular white experience with madness that I find rather problematic. This is also a critique that Pickens specifically advances in her book Black Madness, and since the author draws on Pickens later on I am all the more surprised by this elision here.
I am glad the second part of the piece focuses on a Black author, although Beloved has been analyzed quite a bit from a mad studies perspective -- what about all the other authors of color who have written about madness, autobiographically and otherwise? There would be so many texts that could have been chosen that would have truly added something new because so many authors of color (especially the mad ones) are ignored in mainstream academia. Choosing a different text to analyze here would make this article way more original, but I understand that this piece is so far developed that this would constitute an unrealistic revision. On a final note, I wonder why La Marr Jurelle Bruce's work was not included, since it would expand the author's insights alongside Pickens's -- or Sami Schalk's, or other mad/disability studies scholars of color. I also want to note the problematic absence of Black feminist thought of the last decades that not only has theorized with Morrison's Beloved, but has especially advanced our thinking around trauma (not that the field of trauma studies is always in deep conversation with it).
I think this piece is definitely publishable and I appreciate, again, the care the author has brought to this piece and the texts it engages with.
Author Response
Dear Reviewer One,
I am very grateful for your review and suggestions. I used your comments to re-tool the essay by grounding its argument in the work of La Marr Jurelle Bruce -- I was then able to address and grapple with Sexton's whiteness, and I thank you for bringing this and other issues to my attention.
I have also tried to bring in scholarly discourse and trauma studies work related to Toni Morrison's work to address the issues you raised about the second half of the piece, which you felt was the stronger section. I understand the issue you raised about there being other writers whose work would have helped to demonstrate this theory, but I wanted to show a depth of analysis in two different contexts, and so it made sense for me to try to focus on two works, rather than to bring in more works and not be able to go into depth with them. My hope is that others will see this essay as a starting place for a literary theory of bedlam, and that it will be applied to other works by other writers.
Again, I thank you for your guidance!
Reviewer 2 Report
The author's overall argument for a "making Bedlam" as a practice of reading, writing, and a social justice orientation is an original and exciting intervention. There are moments of powerful prose and argumentation throughout the essay, but the essay would also benefit from significant revision, specifically focusing on more careful parsing of the terminology/concepts deployed, more effective organization, and additional integration of sources. I have included a few overarching suggestions below as well as comments throughout the attached document:
- Intro is rocky--the historical narrative of the history of Bedlam as a term and institution is choppy and relies too heavily on a single source.
- Once the author begins to outline the paper's argument and their conceptualization of "making Bedlam," the essay begins to hit its stride.
- The use of Anzaldua on p. 3 implies that theorizing around mestiza identity can be mapped directly onto theorizing around madness. This analogy raises some problematics, and the author should discuss their choice more explicitly.
- The author at times seems to use the terms "making Bedlam," "mayhem," "madness," and "riot" interchangeably, and at times the author seems to be using them for different purposes. I recommend that they define each term more precisely, explain how each will be used in the context of the argument, and then remain consistent in that use throughout.
- There is very little reference to the critical conversation into which the author is intervening in the paragraphs on pages 2-3. Consider integrating an overview of the critical conversation in Mad studies and then positioning yourself within it.
- There are no trauma studies texts cited in this essay. This is a significant oversight in an essay that is explicitly framed around trauma, making the trauma-related portion of the argument read as imprecise and undertheorized
- Check your use of the phrase "trauma response" and define the way you are deploying it throughout the essay
- The Morrison section is underdeveloped. It implies that madness is explicitly mentioned in the preface to Beloved, but this is not demonstrated. If the author is reading something else (ghosts?) as madness, then they need to spend more time establishing a convincing argument about the connection.
- Reorganize
- Begin the Morrison section with Pickens in order to establish the way you are reading madness into Beloved

Author Response
Dear Reviewer Two,
As I am submitting the revised version of my manuscript, I would like to thank you for your extensive comments, which both percolated in my mind over the past two months and helped me to re-focus the essay.
I took your first suggestion to heart regarding a need to define terminology more closely and deliberately, and I tried to do so, especially in the beginning, most theoretical and foregrounding, section of the essay.
In agreement that the first section relied primarily too much on one source and one kind of source, I reworked the introduction-- I did not get rid of the word origins information, as I think that a history of the word "bedlam" is important to a literary theorization of it, but instead of grounding the theory sort of haphazardly, I strove to newly ground it in the Mad theoretical work of La Marr Jurelle Bruce.
I tried to clarify my reference to Anzaldua's work in response to your concern, so thank you for that, as well.
One of your most helpful critiques was that I had not defined "making bedlam" enough and that I had rendered it synonymous with mayhem while using them differently, perhaps. I decided that I should remove almost all references to mayhem, and, rather than defining mayhem, would focus my efforts on defining bedlam in relation to La Marr Jurelle Bruce's concept of Mad responses to the regime of Reason.
While I did not have the space to be able to do a review of the field of Mad studies in this essay, especially given that it is being published in a literary journal and not a Mad studies journal, I was not able to do an entire review and thought it was not necessary to my purposes, but I did add additional sources, which I hope will address the issues you raised on that matter, although I was referring to many Mad scholars in the original form of the essay, or at least I was under that impression in writing it.
Regarding your criticism about trauma studies: I believe I have addressed it by explicitly stating how I am using the phrase "trauma-informed" and clarifying that this is not a trauma studies/theory essay-- I also incorporated the work of Cathy Caruth and Bessel Van Der Kolk to make reference to trauma studies.
I would argue that Madness is explicitly invoked in the preface to Beloved when Morrison uses the word "sanity" for sanity is a binaristic term that always implied that on the other side is insanity, and I believe I did try to make that clear. I hope that the additions I have made to this section will help to make it more clear how I am using madness in relation to Morrison's work.
Thank you very much for your very thorough feedback!
Round 2
Reviewer 2 Report
The author has done extensive revision on this manuscript, and it is now a much stronger piece. I recommend careful copyediting but no other revisions are necessary at this point.